# Young children show negative emotions after failing to help others

Stella C. Gerdemann[1,2‡*], Jenny Tippmann[3,4‡], Bianca Dietrich[5‡], Jan M. Engelmann[6‡], Robert Hepach[7‡]

1 Department of Early Child Development, Faculty of Education, Leipzig University, Leipzig, Germany, 2 Leipzig Research Center for Early Child Development, Leipzig University, Leipzig, Germany, 3 Institute for Medical Informatics and Biometry, Dresden University of Technology, Dresden, Germany, 4 Center for Evidence-based Healthcare, Dresden University of Technology, Dresden, Germany, 5 Department of Psychology, University of Bremen, Bremen, Germany, 6 Department of Psychology, University of California, Berkeley, Berkeley, United States of America, 7 Department of Experimental Psychology, University of Oxford, Oxford, United Kingdom

‡ SCG, JT and BD are share first authorship on this work.
‡ JME and RH are share senior authorship on this work.
* stella.gerdemann@uni-leipzig.de

**Data Availability Statement:** Both studies were preregistered, and all data and analysis scripts associated with this study have been made available through the Open Science Framework (https://osf.io/uvtch/).

## Abstract

Self-conscious emotions, such as guilt and shame, motivate the adherence to social norms, including to norms for prosociality. The relevance of an observing audience to the expression of negative self-conscious emotions remains poorly understood. Here, in two studies, we investigated the influence of being observed on 4- to 5-year-old children's ($N$ = 161) emotional response after failing to help someone in need and after failing to complete their own goal. As an index of children's emotional response, we recorded the change in children's upper body posture using a motion depth sensor imaging camera. Failing to help others lowered children's upper body posture regardless of whether children were observed by an audience or not. Children's emotional response was similar when they failed to help and when they failed to complete their own goal. In Study 2, 5-year-olds showed a greater decrease in upper body posture than 4-year-olds. Our findings suggest that being observed is not a necessary condition for young children to express a negative self-conscious emotion after failing to help or after failing to complete their own goal. We conclude that 5-year-olds, more so that 4-year-olds, show negative emotions when they fail to adhere to social norms for prosociality.

## Introduction

Self-conscious emotions, such as shame and guilt, motivate people's adherence to social norms and standards, including to norms for prosocial behavior [1, 2]. Consider the emotion shame. Although shame is a highly aversive emotion [3–5], it serves important social functions. For instance, a lack of shame following social transgressions (e.g., stealing, cheating) can lead to ostracism or, in the worst-case, expulsion from ones' social group [6]. Further, the anticipation of shame [or guilt; 7] can motivate prosocial actions and inhibit anti-social tendencies, and thereby contribute to the stability of cooperation [1, 2, 6]. Both shame and guilt are thought to

**Funding:** Research for this project was partly funded by a 'Flexible Fund' grant awarded to R.H. by Leipzig University. S.C.G. was supported by a doctoral grant of Leipzig University ("Doktorandenförderplatz"). There was no additional external funding received for this study.

**Competing interests:** The authors have declared that no competing interests exist.

result from self-reflection and a negative self-evaluation following a transgression or a failure to meet a (social) standard [8]. Finally, guilt leads to efforts to repair the caused damage via reparative prosocial behavior [e.g., 9, 10; see also 11 for evidence that shame has a similar function].

In the current investigation, our aims were (a) to explore whether young children express a negative self-conscious emotion, similar to shame or guilt, after failing to help others. In addition, (b) we aimed to assess whether children's emotional response is influenced by being observed, and (c) whether children show similar self-conscious emotions in response to failing to help and after failing to achieve their own goals.

## Failures to help and young children's self-conscious emotions

Our first goal was to examine when and whether children express negative self-conscious emotions after failing to help others. Although children begin to help others early in life [e.g., 12], no study to our knowledge has investigated children's emotions in response to failing to help others [13]. Prior investigations have focused on children's positive emotions in response to successfully helping others [14–18]. These studies have found that 2- to 5-year-old children express more positive emotions (as assessed by coders) in response to giving resources to a puppet than when receiving resources for themselves [14, 15, 18]. Moreover, 2-year-old children express an elevated upper body posture after helping to fulfill an adults' needs [17]. In a separate line of work, using the so-called mishap paradigm, toddlers are led to believe that they caused harm to someone by accidentally breaking their doll or tower. In these studies, 2- to 4-year-old children have been found to express rudimentary forms of guilt and shame—indicated, for instance, via an averted gaze, a shrunken or tense posture, an avoidance of the person harmed, as well as through apologies and repairing the caused damage [19–22]. Some scholars have, furthermore, argued that young children's emotional responses in the mishap paradigm can be interpreted as either shame-like (characterized by avoidance the person harmed, bodily expressions of tension [e.g., hunching ones' posture], as well as gaze aversions) or guilt-like [characterized by a reparation of the casued damage and apologies; 19, 20, 23]. However, other scholars have doubted whether this disticntion between shame and guilt can be drawn in early childhood, because children's early self-conscious emotions may not yet be distinct in this way [21]. Moreover, some scholars have found evidence for guilt- and shame-like responding as early as age 2 [19, 20], while other scholars have found that 3-year-olds express more distinctly guilt-like responseses than 2-year-olds [22]. These emotional responses appear to persist throughout preschool age [21, 24], although only few studies have examined the expression of shame and guilt over a larger developmental period [e.g., 21]. In sum, prior investigations suggest that sometime during late toddlerhood and early preschool age, young children begin to appreciate the norms and standards of their social group, and evaluate their own behavior from that perspective resulting in the expression of shame and guilt when children behave poorly or violate social standards [e.g., by breaking someone's belongings; see 25].

A further line of research has found that children express negative (shame-like) emotions after failing to complete their own goals, for instance after failing to build a tower for themselves or after failing to throw a ball into a hoop [26–30]. In these settings, children between 2 to 5 years of age appear to establish a standard (e.g., of completing a goal, such as finishing a puzzle or tower for themselves), and express negative emotions, presumably indicative of a negative self-evaluation, when they fail to meet that achievement-related standard [30]. It remains less clear whether and when young children show emotions in response to failing to help others that are similar to the emotions that they express after harming others [19–22] or after failing to complete tasks for themselves [26–30]. Evidence for such an emotional response would suggest that young children view helping others as required.

There is already some evidence that preschoolers possess normative standards regarding some kinds of prosocial behaviors. For instance, by age 5 to 6 (but not age 3 to 4) children protest third parties' failures to be charitable towards poor individuals [31]. Similarly, when explictly judging others' helping in hypothetical scenarios young children by age 3 evaluate helping as obligatory in most contexts [32]. In sum, preschoolers' verbal judgements and protest of others' (failures to) help suggests that young children possess a nascent sense that helping is prescribed by social norms. It remains poorly understood whether and when young children experience helping others as personally binding to the extent that not helping others causes a negative emotional response.

## The influence of observation on the expression of negative self-conscious emotions

Our second goal was to examine whether children's expression of negative emotions following a violation of a social standard depends on being observed. Emotions can have both intra- and interpersonal functions [33, 34]. That is, emotions can influence behaviors as both feelings and as emotion displays that signal information to observers. For instance, the nonverbal expression of shame, including a hunched posture and an averted gaze, has been argued to function as an appeasement display that communicates the expressers' negative self-evaluation to observers following a transgression [3, 35, 36]. Moreover, children are more likely to forgive transgressors who express remorse (including through apologies) after harming others [37].

While children have been found to be more forgiving of remorseful transgressors [37], it is unclear whether children themselves are more likely to express a negative self-conscious emotion when they are being watched, and their emotion display could hence influence others' judgement of themselves. In one study, Holodynski [28] found that preschool children between 3 to 6 years of age are more likely to show shame (according to the author's definition) when they fail a task while they are observed than while they are alone. In this study, children were tasked with completing a set of increasingly difficult puzzles and were either asked to show the experimenter how many puzzles they could complete or complete the puzzles on their own, while the experimenter was outside of the room. Children in this study were said to express shame in response to a failure to complete a puzzle if they showed three or more shame-relevant emotion features (e.g., a shrunken body posture, a negative verbal self-evaluation [e.g., saying "I can't do this"], an averted gaze, looks away from the observer). In Holodynski [28] children were found to express shame more often while they were in the presence of the experimenter who assigned them with the task to complete than while they were alone.

In addition, Harter [38] reported the results of interview-studies showing that children at age 5 to 6 only mentioned whether a parent was ashamed of them in response to hypothetical situations in which they failed a task (during an athletic competition). By age 6 to 7, children reported shame only while their parent was watching an achievement-related failure. Finally, by age 7, children reported shame in response to achievement-related failures independent of observation by their parent. In sum, one line of work supports the notion that children express more shame when they are being observed by others (e.g., their parents or an experimenter). According to this view, the tendency to focus on others' judgement of oneself following failures decreases with age during early childhood and is replaced by intrinsic self-evaluations that emerge during school-age.

It is important to note that not all scholars agree that young children's expression of self-conscious emotions is motivated by concerns with how their behavior is evaluated by others [e.g., 30, 39; see also 4 for similar results from adults]. For instance, Stipek et al. [30] found that 2- to 5-year-old children express pride in response to success with only a modest influence of

being praised by an adult on children's emotional response. This finding led the authors [30] to conclude that young children's self-conscious emotions result from intrinsic self-evaluations that are independent of adults' evaluation of their behavior during the preschool years. Although Stipek et al. [30] also investigated children's shame-like emotions in response to failures, showing that 2- to 5-year-old children express negative emotions when they fail to achieve their own goals, the authors did not examine whether adult's evaluation of their behavior influences children's emotional response to failures.

Several aspects limit the conclusions that can be drawn from prior work on the influence of observation on children's expression of shame. For instance, in Holodynski [28] children were observed by the same experimenter who provided children with a task to complete. If children expressed a more shame after failing to complete a task in the presence of the experimenter, who assigned them a task, children's emotional response might be due to fear of being reprimanded by the adult in response to failing the task. Therefore, it remains unclear, whether young children are more likely to express negative self-conscious emotions in the presence of an uninvolved third party. Such an emotional response could serve to appease onlookers [e.g., 36] and make them more forgiving of the expresser following her transgression or failure to meet a social standard [e.g., 37]. Thus, expressing a negative self-conscious emotion in the presence of observers (compared to when nobody is watching) could function to enhance young children's reputation. In other settings, young children have been found to be concerned with their reputation. For instance, by age 5, young children behave more generously and steal less when they are being watched compared to when their behavior is anonymous [e.g., 40, 41; see 42 for a review].

In addition, Holodynski [28] relied on both verbal and nonverbal expressions to identify shame in young children. The author's coding system to identify the expression of shame included features such as a negative verbal self-evaluation and looks away from the observer. Such (verbal) emotion features of shame may necessarily occur more often in the presence of observers because they are clearly communicative. Therefore, it remains poorly understood whether young children's nonverbal expression of a self-conscious emotion following a transgression is more pronounced when children are observed. Finally, no investigation to our knowledge has assessed whether observation influences young children's expression of self-conscious emotions after failing to help, that is after failing to meet a prosocial standard. There is mounting evidence that preschool children attempt to manage their *prosocial* reputation by behaving more generously and less antisocially while they are being observed compared to unobserved settings [e.g., 40, 41]. By contrast, little is known regarding whether children use their emotional expression to enhance their reputation after failing to cooperate.

## Children's emotional responses to failures to help and failures to complete their own goals

Our third goal was to explore whether young children express similar or different emotions when they fail to help and when they fail to complete their own goals. We included an own-goal context primarily as a comparison condition to the novel help context, as children have already been found to express negative emotions when they fail to achieve their own goals [e.g., after they fail to complete their own tower or puzzle for themselves; 26–30]. In addition, previous studies have shown that 2-year-old children express less emotional arousal when they see an action remains unfulfilled (in response to a scene that needs to be "cleaned up") compared to a scene in which someone's need remains unfulfilled [43, 44]. Therefore, we predicted that children would express a more negative emotion after failing to help others, because the helpee's need is still unfulfilled, than in response to a failure to complete their own goal.

## The current studies

In the present studies, we explored the development of children's emotional response after not being able to help an adult to complete a task. The design of our task was based on a previous study in which children expressed shame after failing to complete their own goals [29].

There is currently a lack of methods to measure young children's emotions automatically and objectively. Prior studies have used behavioral coding and observation to assess 2- to 5-year-old children's emotional expression following a transgression or a failure to meet a standard [19–22, 26–30]. In the current investigation, we primarily assessed children's nonverbal emotional response using a motion depth sensor imaging camera, the Microsoft *Kinect* [17, 45, 46]. The *Kinect* enables the unobtrusive and objective quantification of changes in children's upper body posture based on an automated video analysis. Young children's expression of self-conscious emotions, such as shame and guilt, has been characterized by features such as a lowered, hunched or tense body posture, and slumped shoulders, as well as an overall negative emotional expression [19–21, 28–30]. Therefore, our methodological approach allowed us to precisely quantify one aspect of children's nonverbal expression of a negative self-conscious emotion: their lowered upper body posture.

We note that some authors have argued that a lowered body posture is a feature that is predominantly shame-relevant [3, 19, 20, 23, 29], while guilt has been argued to be predominantly expressed via reparative prosocial behavior, as well as confessions of the harm caused, including apologies [3, 9, 10, 19, 20, 23]. At present, our investigation does not speak to the question of whether shame and guilt are expressed in this distinct way in young children, as we focused primarily on children's nonverbal emotional expression. Therefore, we follow previous authors [e.g., 21], in asking whether children show evidence of a negative self-conscious emotion without attempting to objectively distinguish between the expression of shame and guilt.

In Study 1, we compared 5-year-old children's emotional response in a prosocial and achievement-related context, by varying whether children were unable to fulfil their partner's goal (help context) or their own goal (own-goal context). Children were tested in an observed or an unobserved condition. Thus, Study 1 had a 2x2-design with the factors goal context (help or own-goal) and observation (observed or unobserved). We predicted (a) that children would express a negative emotion (lowered upper body posture) after failing to help. In addition, (b) we predicted that children would show a more negative emotion (lowered upper body posture) in the observed compared to the unobserved condition. This prediction was based on the prior evidence from both adults and children suggesting that observation influences the expression of shame-like emotions [3, 28, 35, 36, 38, 47; although see also 4, 30]. Third, (c) we predicted that children would show a more negative emotion (lowered upper body posture) in the help compared to the own-goal context.

In sum, Study 1 addressed three research questions:

1. Do 5-year-old children express a lowered upper body posture in response to not being able to help a needy social partner?

2. Do 5-year-old children express a greater reduction in upper body posture after failing to help in an observed compared to an unobserved context?

3. Do 5-year-old children express a greater reduction in upper body posture in response to not being able help than after not being able to complete their own goal?

In a Study 2, we addressed the question whether 4- to 5-year-old children express a lower upper body posture in response to failing to help, and whether this emotional response is influenced by being observed.

## General method

Both Studies 1 and 2 were preregistered at the OSF (https://osf.io/uvtch/). These studies received were approved (in writing) by the ethics review board of the medical faculty of Leipzig University (IRB number: 169/17) and were carried out in a midsized city in Germany. Participants were recruited from a database of children with mixed socio-demographic background and testing took place at children's preschools. Parents gave written informed consent for their child's participation.

### Body posture recording, pre-processing, and measures

In both studies, children's body posture was recorded twice during a baseline phase (before the experimental manipulations) and twice during a test phase (after the experimental manipulations, i.e., after children failed to help or complete their own goal) using a Microsoft Kinect camera, which was operated using a script run in Matlab (Version 9.5). The Kinect provides the location of x-, y- and z-coordinates of 20 skeletal points for each recorded sequence (see Fig 1; note that the individuals depicted in this figure or their legal guardians gave written informed consent as outlined in the PLoS consent form for publication of this figure).

The processed data comprised body posture data along 20 increments of children's distance from the Kinect camera (see https://osf.io/uvtch/ for preprocessing steps). Our analysis focused on 13 increments, for which children provided more than 90% of the median number

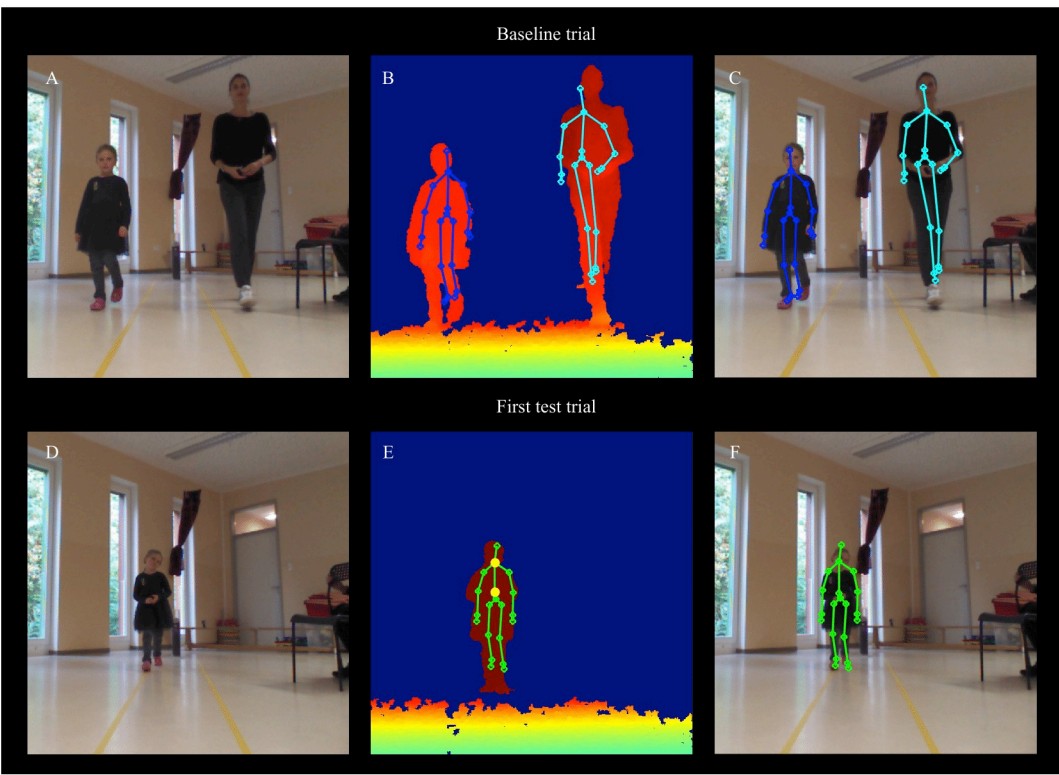

**Fig 1. Examples of the data provided by the Kinect.** Top row: baseline trial; bottom row: first test trial. (A, D) Color images of a baseline trial and first test trial respectively. (B, E) Depth images with skeletal points mapped onto them. (C, F) merged images (skeletal points mapped onto the color images). Yellow circles on (E) represent the chest and hip center data points used for the analyses. Analyses focused on the child's change in body posture. The individuals depicted in this figure or their legal guardians gave written informed consent (as outlined in the PLoS consent form) for publication of this figure.

of data points. This approach parallels previous applications of body posture analyses [17, 46]. For the analyses, the data provided during the baseline phase was averaged across the two trials.

**Measures.** We report the results of three body posture measures:

1. The change in chest height (the baseline-corrected y-value of the chest center data point)

2. The change in hip height (the baseline-corrected y-value of the hip center data point)

3. The change in chest expansion (the result of subtracting children's change in hip height from their change in chest height, see Fig 1).

Previous studies have generally found that the change in children's chest height reflects the valence of children's emotional expression [17, 46]. For instance, patterns seen for the change in children's chest height were found to parallel patterns for the number of smiles children showed, as well as their general affect—as assessed by adult coders. We had planned to examine children's change in chest height as an indication of children's emotional response and conduct a control analysis on the change in children's hip height, like in previous applications of body posture analyses [17, 45, 46]. In our studies, however, unlike in these past investigations, we observed substantial variation in children's change in *hip* height across experimental manipulations, which limits the interpretability of children's change in chest height. If children's hip height varies substantially, we cannot confidently conclude that the patterns seen for children's change in chest height are not due to running, jumping, or crouching. Therefore, here, we analyzed a corrected measure of children's change in chest height, which resulted from subtracting children's change in hip height from their change in chest height (= children's change in chest expansion). Through additional emotion coding our investigation corroborated that the change in children's chest expansion is a robust measure of the valence of children's emotional expression (see **Emotion Valence Coding** and **Discrete Emotion Coding** below). For space reasons, we present the analysis of children's change in chest expansion in our main manuscript and describe details of children's change in chest height and change in hip height elsewhere (in S4 and S8 Appendices). The analysis of children's change in chest expansion should, however, be considered exploratory, as it was not part of our original preregistration.

## Study 1

### Participants

The final sample of children who provided body posture data for Study 1 included *N* = 68 preschoolers (mean age: 5 years, 6 months, 30 days; standard deviation: 4 months, 9 days; range: 5 years, 0 months, 3 days to 6 years, 2 months, 2 days, 34 boys), with an approximately equal number of children of each gender in each condition. Additional *N* = 37 children were invited to participate in the study, yet their data was not included in the analyses (see S2 Appendix, Tables A and B). Children's data was not included if children became upset during the study (i.e., if children started to cry or said that they wanted to return to their Kindergarten group; *N* = 6); due to an error in the apparatus that allowed children to access the object they needed to successfully help or complete their own goal (*N* = 3), or if children did not attempt to help or complete their own goal, by failing to interact with the tube apparatus (*N* = 13). The decision to include children only if they attempted to help or complete their own goal is comparable to previous studies [48]. In addition, as preregistered, if pre-processing steps (see https://osf.io/uvtch/) did not result in any usable data on the first test trial, which was recorded immediately after children's failed attempt to help, the child's data was excluded from all further

analyses (*N* = 15). Fifty-seven out of 68 children provided data for a second test trial, again approximately equally distributed across conditions.

## Design and materials

Study 1 had 4 conditions, which were crossed in a 2 (observation: observed or unobserved) x 2 (goal context: help or own-goal) between-subjects design. Children were randomly assigned to experimental conditions. Children provided body posture data during a baseline phase and a test phase, which each included two trials. Our hypotheses regarding the influence of goal context and observation focused on the first test trial.

We, in addition, compared children's upper body posture from the moment after they failed to help (Trial 1) to the resolution of the situation (Trial 2), to assess whether children's upper body posture after failing to help or complete their own goal was lowered after failing to complete their own or the experimenter's goal compared to the resolution of the situation. Comparing children's emotional response across different phases of an experimental paradigm is similar to the approach taken in prior work on children's expression of positive emotions in response to successfully helping others [14].

Materials included a child-sized table, wooden pieces to build a tower, and an image of the finished tower. In addition, there was a plexiglass tube with a special tower block inside referred to as the tower's "crown" (see Fig 2). The tube was manipulated so that it was impossible to reach the crown if a plexiglass slider blocked access to it. Pieces to build a simple chair were placed on one side of the room in the unobserved condition, and in the observed

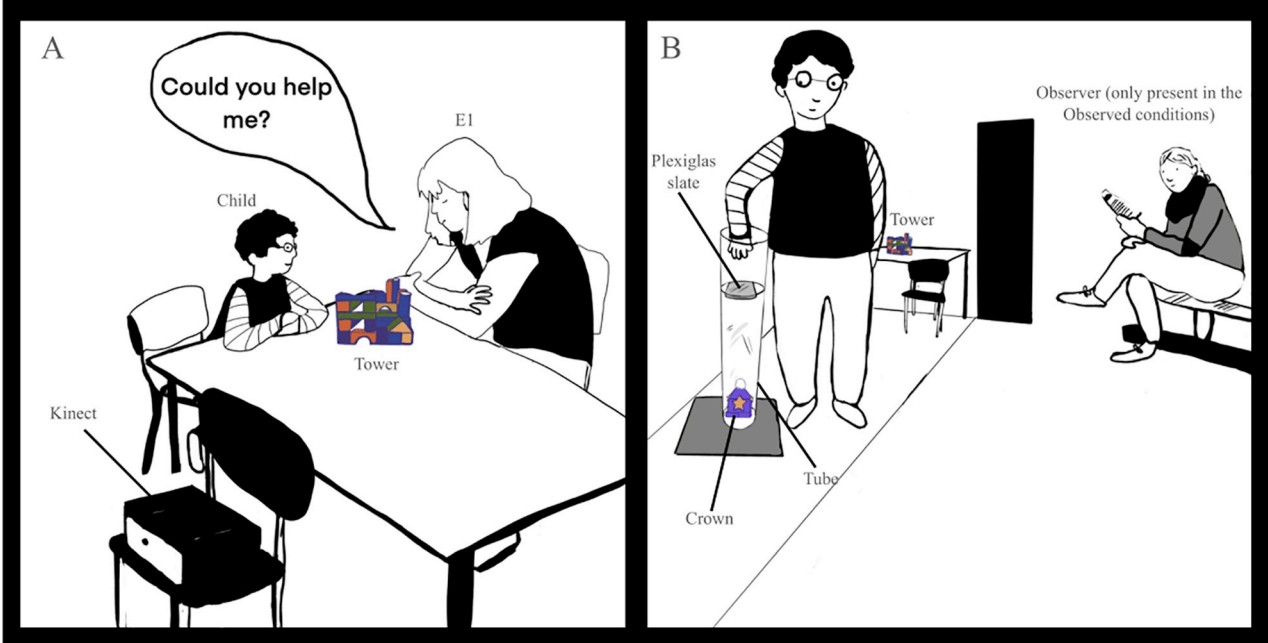

**Fig 2. An illustration of key phases of the procedure.** (A) In the help goal context, E1 asks children to help complete her tower by retrieving the crown from the tube while, in the own-goal context, E1 tells children to complete their own tower. In both conditions, children see a picture of the completed tower, and are told that the crown is needed for the tower to be completed. (B) After asking children to help or complete their own goal, children attempt, but fail, to retrieve the crown from the tube because the tube is blocked by a slider. Depending on condition, children are either observed by E3 or alone. After the failed attempt to help or complete their own goal, children's body posture is measured for a first test trial by the Kinect camera, as they walk towards the door. The illustration shows the tube apparatus used in Study 1.

condition the observer was occupied with building the chair. The *Kinect* camera was concealed by a cardboard box and placed so that children's body posture could be recorded.

## Procedure

Children interacted with a main experimenter (E1) at the child-sized table in a quiet room of their day-care center. A second experimenter (E2) operated a laptop, which controlled the *Kinect* camera from outside of the room. In the observed conditions, a third experimenter (E3) took on the role of the observer.

**Baseline phase.** In all conditions, children were engaged in building a tower with E1 at the study table. Children were shown an image of the tower and told that it should look like the picture by the end of building it. The observed and unobserved conditions differed with regards to the presence of E3. In the observed conditions, to provide a cover for the presence of the observer (E3), E1 introduced E3 as a "craftswoman/man" and told children that E3 would "watch them today". In the observed conditions, E3 was occupied with building a chair in one corner of the room. Explicitly telling children whether they can be observed is comparable to other studies, that have elicited effects of observation on children's prosocial behavior [40, 41]. E1 explained that the tower's "crown," a special piece on top of the tower, would have to be added to the tower for it to be completed. In the help context, E1 expressed her desire for the tower to be completed and indicated that she would be sad should the tower remain incomplete. For instance, she said: "Look, my crown *(points to image)* is covered in purple paper and has yellow stars on it. I made it myself, and I'm really looking forward to adding it to my tower in the end. *(In a sad voice)* But I would be sad if the crown weren't added to my tower later." In the own-goal context, E1 referred to the tower as the child's tower, and children were merely told that their tower would be incomplete if they did not add the crown. For instance, she said: "Look, the crown *(points to image)* is covered in purple paper and has yellow stars on it. It must be placed on your tower at the end, so that it is completed. Because, without the crown, your tower will not be finished." Once all pieces of the tower on the study table had been added to the tower, children and E1 retrieved two missing pieces (one at a time) for the tower from the foot of the tube (baseline recordings 1 & 2). While children walked toward the study table, two successive baseline recordings of the child's body posture were taken with the *Kinect* camera. At this point, the tower was only missing its "crown".

**Test phase.** E1 then pretended to notice that she urgently had to leave. In the observed conditions, children were reminded that E3 (the observer) would remain in the room. In the help context, E1 said *(in a sad voice and while showing a sad facial expression and posture)*: "Oh no, my tower, it's not completed yet. It's still missing my hand-made crown! I would be so sad if my tower weren't completed with my crown. Will you help me? Will you retrieve my crown from the tube, and add it to my tower?" In the own-goal context, on the other hand, E1 said *(in a neutral voice and with a neutral facial expression and posture)*: "Oh no, your tower! It's not completed yet. It's still missing the crown on top. Your tower wouldn't be completed without your crown. Will you finish your tower? Will you retrieve your crown from the tube, and add it to your tower?" If children agreed to complete their/E1's tower, E1 told children to walk towards the door once they heard a knock on the door, and that they would have to return to their day-care group. E1 then left the study room. By asking children to complete their tower in the own-goal context, E1 transferred the goal of completing the tower to the child. Our help manipulation was similar to that of previous studies which examined children's willingness to fulfill emotional needs [49].

E1 then waited outside of the study room, while children attempted to retrieve the "crown" from the tube. After 1 minute elapsed, E1 knocked, which prompted children to walk towards

the door (first test trial). While children walked towards the door, data for the first test trial was recorded with the *Kinect*. The choice of a 1-minute duration for children's attempt to help was the result of piloting the procedure. If children returned to the door before 1 minute elapsed, their body posture data was nonetheless recorded, and children were nonetheless included in the sample. Additional coding revealed that children spent an approximately equal amount of time attempting to retrieve the crown in all four conditions (see S5 Appendix). Note that children who succeeded in helping or completing their own goal were excluded from the sample (see **Participants**). E1 then returned to the study room and told children they were unable to retrieve the crown, because of a mistake by E1. E1 then removed the slider from the tube and handed children the crown. Children then walked towards the study table with the tower and could add the crown to the tower thereby finishing the task they were assigned or helping the experimenter (second test trial). During this "resolution" of the situation, while children carried the crown to the tower, a second test trial was recorded using the Kinect. In line with previous investigations of children's self-conscious emotions [e.g., 19], children did not intentionally refuse to help E1 or complete their own goal.

## Coding and analyses

**Emotion valence coding.**   We conducted a supplementary emotion valence coding to corroborate the validity of our body posture measures. Importantly, the body posture measures have already been partly validated as measures of emotion valence in prior work [17, 45, 46]. However, given the novelty of our task, as well as of the exact measure (the change in children's chest expansion as opposed to the change in children's chest height), our aim was to provide further data to bear on the question whether the change in children's upper body posture reflects the valence of children's emotional expression. Two coders independently rated the valence of children's emotional expression during the baseline phase, as well as on the first and second test trial on a scale from -4 (significantly negative) to +4 (significantly positive; see also 17 for a similar coding procedure). Ratings were conducted based on video stills (without audio), which corresponded exactly to those that were used to extract children's body posture data (see Fig 1D). Coders were instructed to look at all available video frames of a respective trial and base their emotion valence rating on the entire recording. The emotion valence ratings for the first and second baseline trial were correlated with each other, $r = .55$. Therefore, the two ratings were averaged to create one baseline rating per child and coder. Across the baseline phase, and both test trials, inter-rater agreement was as follows: $ICC = .58$ ($r = .49$). Ratings were averaged across the two coders within each phase for the analyses.

**Body posture analysis.**   We tested our predictions regarding the change in children's body posture using linear mixed models [LMMs; 50]. As preregistered, we analyzed the change in children's body posture on the first test trial to test our predictions regarding the influence of observation and goal context. Children were only observed or unobserved on the first trial, therefore only the first trial is informative regarding these research questions. The first trial models included random effects for participant, kindergarten (indicating which daycare center children attended, dummy coded), and time-distance on participant. The variable time-distance indicates children's distance (standardized) from the *Kinect*.

To explore, in addition, whether children's body posture differed from the moment after failing to help or complete their own goal (Trial 1) to the resolution of the situation (Trial 2), we ran omnibus models that included data from both test trials. The omnibus model included the same fixed and random effects of the first trial model, and an additional fixed effect of trial and random slope of trial on participant.

Models were computed in RStudio [Version 1.2.1335, 51], using the package *lme4* [Version 1.1–23, 52]. Significance of the individual fixed effects was tested using likelihood-ratio tests comparing the full model with reduced models without the fixed effect of interest [53]. The analysis strategy of fitting two models, one including the three-way-interaction with time-distance and one including only two-way interactions parallels previous work using the same technology [17].

## Results

### Body posture

**Change in chest expansion.** *First trial analysis.* Children's chest expansion (in cm) reliably decreased from baseline to the first test trial across all four conditions: help observed, 95% Confidence Interval (CI) [-1.68, -5.26]; help unobserved, 95% CI [-1.22, -4.94]; own-goal observed, 95% CI [-0.94, -5.49]; own goal unobserved, 95% CI [-1.72, -6.40] (see Fig 3). The change in children's chest expansion, however, did not systematically vary as a function of the two-way interaction of observation and goal context, $\chi^2(1) = 0$, $p = 1$ (see Fig 3). There was also no three-way interaction of observation, goal context and time-distance, $\chi^2(1) = 0.06$, $p = .8$. We did find that boys ($M = -4.47$, $SD = 4.74$) showed a greater decrease in chest expansion (in cm) than girls ($M = -2.46$, $SD = 2.84$), $\chi^2(1) = 4.09$, $p = .04$.

*Omnibus analysis (first and second trial).* In the omnibus model predicting the change in children's chest expansion, there was again an influence of gender, $\chi^2(1) = 6.92$, $p = .01$. Averaging across both test trials, boys ($M = -3.65$, $SD = 4.32$) showed a greater decrease in chest expansion (in cm) than girls ($M = -1.62$, $SD = 3.13$). In addition, there was a clear effect of trial, $\chi^2(1) = 10.45$, $p = .001$. Children's chest expansion (in cm) was lower on the first trial, after children failed to help or complete their own goal ($M = -3.46$, $SD = 4.01$), than on the second test trial, during the resolution of the situation ($M = -1.58$, $SD = 3.48$). There was no

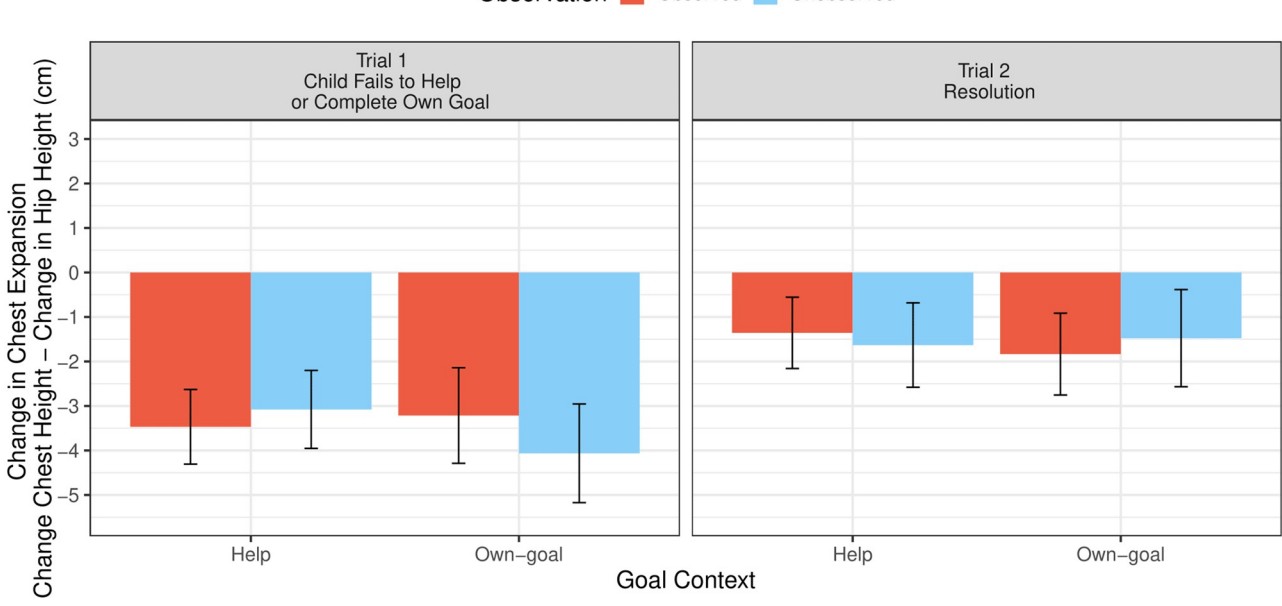

**Fig 3. Children's baseline-corrected upper body posture on the first and second test trial of Study 1.** Values are averaged across increments of time-distance. Bars represent means and error bars represent ± 1 standard error.

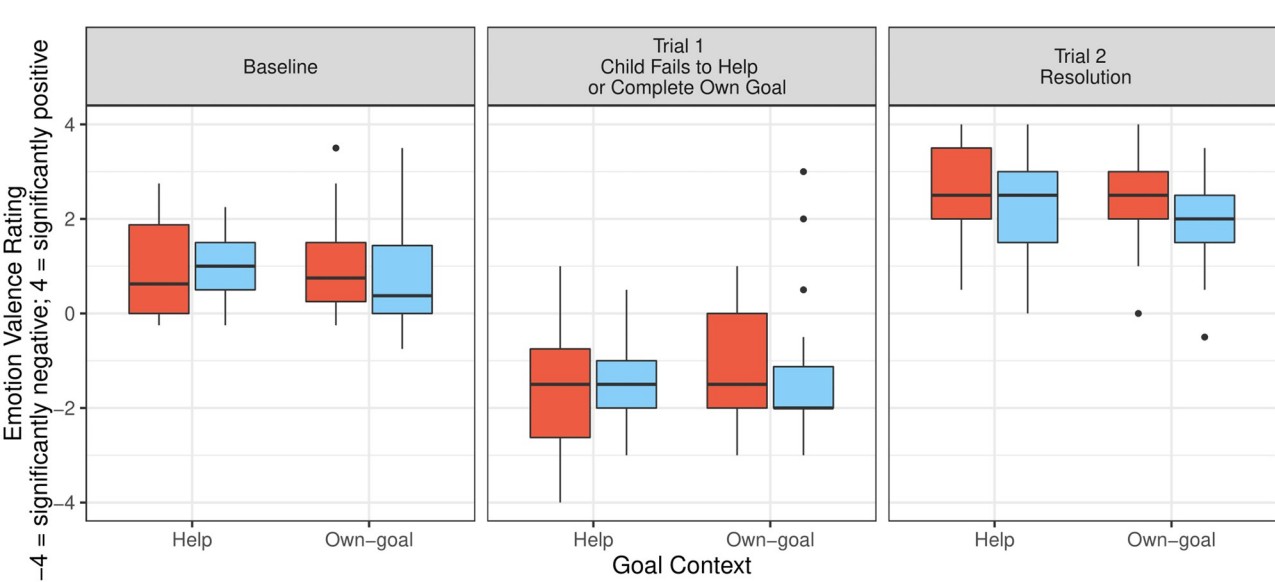

**Fig 4. Boxplot of children's rated emotion valence during the baseline phase, and on the first and second test trial of Study 1.** The videos used to code children's emotion valence are the same ones that were used to automatically extract children's body posture data (see for example Fig 1D). The black lines inside the boxes represent medians. The lines above and below the median mark the first and fourth quartile. The whiskers capture extreme observations and black dots represent observations that are 1.5 times the interquartile smaller than the first quartile or greater than the fourth quartile.

influence of observation or goal context on children's change in chest expansion in the omnibus model (see Fig 3).

**Emotion valence coding.** The analysis of children's emotion valence (assessed by raters) showed only an overall influence of phase, $F(2, 186) = 169.13$, $p < .001$, on children's emotion valence ratings. Children's emotion valence was rated as more negative on the first trial (after children's failed attempt to help or complete their own goal; $M = -1.34$, $SD = 1.29$), relative to the baseline phase (averaged across two trials), while children were neutrally walking to retrieve missing blocks for the tower ($M = 0.96$, $SD = 0.97$), $\beta \pm SE = -2.29 \pm 0.19$, $t(186) = -11.79$, $p < .001$, and relative to the second test trial ($M = 2.35$, $SD = 1.09$), $\beta \pm SE = -3.68 \pm 0.2$, $t(186) = -18.03$, $p < .001$ (see Fig 4). There was no additional influence of goal context, observation, or gender on children's emotion valence, as assessed by coders.

## Discussion of Study 1

Regarding our first research question our findings suggest that children's upper body posture was indeed lowered in response to not helping a needy social partner. This finding is supported by the comparison across trials: children expressed a greater reduction in upper body posture on the first test trial—immediately after children were unable to help—compared to resolution of the situation, on the second test trial. In addition, children's upper body posture was reliably below baseline on the first test trial.

Furthermore, the emotion valence coding revealed, consistent with the pattern for children's change in chest expansion, that children's emotional expression was rated as more negative on the first trial compared to the second test trial and compared to the baseline phase. This coding thus provides corroboration for our automatically recorded measure of the change in children's upper body posture—children's change in chest expansion—as an index of children's emotional response [17, 45, 46].

In contrast to our second prediction that children would express a greater reduction in upper body posture when they are observed compared to when they are unobserved following a failure to help, children's emotional response was similarly negative in both in the observed and unobserved condition. In past work showing an influence of observation on children's expression of shame, both verbal and nonverbal emotion cues were measured [28]. By contrast, here, we focused on children's nonverbal emotional expression.

Finally, our results also do not suggest that children show a greater reduction in upper body posture when children are unable to help compared to when they are unable to complete their own goal. Children may not have expressed a more negative emotion in the help compared to the own-goal context, because children perceived failing to complete their goal of building their tower as a similar failure to meet a (social) standard to failing to help. This suggests that children were not more motivated to complete the goal of the experimenter, and thereby fulfill her needs, but showed a comparable motivation to complete their own goal, presumably out of an achievement-related motive.

## Study 2

In Study 2 we aimed to replicate the findings of Study 1 and examine the development of children's emotional response to a failure to help across a wider age range. Study 2 included 4- and 5-year-old children and focused on two conditions: help observed and help unobserved. Our main research question for Study 2, like in Study 1, was whether children would express a greater reduction in upper body posture after failing to help in an observed compared to an unobserved condition. In addition, we explored whether age or the interaction of age and being observed would influence the reduction in children's body posture. In sum, our research questions for Study 2 were the following:

1. Do 4- and 5-year-old children show a decrease in upper body posture in response to not helping a needy social partner?

2. Do 4- and/or 5-year-old children express a greater decrease in upper body posture when they are observed compared to when they are unobserved during a failure to help?

We found weak effects of being observed on children's change in chest height (uncorrected for the change in children's hip height) in Study 1 (see S4 Appendix). Thus, our primary motive for conducting Study 2 was to address the question whether the results for 5-year-old children's change in chest height in response to failing to help in Study 1 could be replicated. We included a wider age range of children in Study 2 to explore whether observation would influence younger children's emotional response to failing to help as well. The full details of the analysis of children's change in chest height can be found in the Supplementary Information.

Study 2, in addition, allows us to address the question whether children below age 5 might be more sensitive to being observed after failing to help. Such a developmental pattern would be consistent with previous proposals according to which self-conscious emotions are expressed more independently of observation or evaluation through others with increasing age [28, 38]. According to this hypothesis, younger children's self-conscious emotional expression should be more affected by being observed than that of older children.

### Participants

Participants for Study 2 were $N$ = 93 preschoolers. The sample included $N$ = 48 4-year-olds (mean age: 4 years, 7 months, 4 days; standard deviation: 3 months, 10 days; range: 4 years, 0 months, 5 days to 5 years, 0 months, 0 days; 27 boys) and $N$ = 45 5-year-olds (mean age: 5

years, 4 months, 30 days, standard deviation: 3 months, 16 days, range: 5 years, 0 months, 4 days to 5 years, 11 months, 4 days; 20 boys). We preregistered our aim to include data from $N = 104$ children in our statistical analyses, yet altered the stopping rule for data collection, because of the high attrition rate after collecting data from one third of the sample. Our new stopping rule was that we would test $N = 104$ children and replace all children whose data could not be included in the sample (see exclusion criteria). We opted to terminate data collection after replacing those children of the initial sample of $N = 104$ who did not provide data for the analyses. Our final sample size was not much smaller than the one we preregistered, because minor improvements to the study set-up (e.g., placing the Kinect at the right angle), enabled an improved recording of children's body posture, resulting in a lower rate of data exclusion due to data loss at the end of data collection.

A total of $N = 59$ additional children were recruited to participate, yet their data was not included in the analyses (see Table A and B in S6 Appendix). Like in Study 1, data was excluded, if children became upset during the study ($N = 9$); if children did not attempt to help, by failing to interact with the tube apparatus while E1 was outside of the room ($N = 9$), and due to an error in the apparatus that allowed children to access the object they needed to help ($N = 1$). Data from one additional child was excluded because of an interruption of the study during the test phase ($N = 1$). As preregistered, data from $N = 39$ children was excluded because pre-processing steps did not result in any usable data on the first test trial or on all baseline trials (see https://osf.io/uvtch/ for pre-processing steps). This rate of exclusion (~26% of the entire sample) due to data loss, is comparable to the data loss of other developmental work using similar physiological methods [e.g., neurophysiological measures, 54]. Seventy-five children provided data for a second test trial.

To determine sample size, we ran a priori power analyses by simulating the models for Study 2 using the R-package *simr* [55]. These analyses revealed above .8 power to detect the main effects and two- or three-way interactions of interest with $N = 104$, $\alpha = 0.05$, based on the effect size of the interaction effect seen for children's change in chest height in Study 1 (Estimate = -0.031). Given that our final sample size was slightly smaller than anticipated, we conducted sensitivity power analyses using the same effect size. These analyses estimated average power between .75 to .76 to detect two- and three-way interaction effects and .97 power to detect main effects given our final sample size.

## Design and materials

Study 2 had two conditions (help observed and help unobserved) which were tested in a between-subjects design. Children were randomly assigned to conditions. Like in Study 1, in each condition, children provided data during a baseline phase, and during a test phase, which each included two trials. The materials in Study 2 were nearly identical to those used in Study 1, with one small change to the apparatus, with which we aimed to make helping look easy enough, so that younger children would have the expectation that they could succeed at helping. Past work has shown that for children to show negative shame-like emotions in response to failures, the task children attempt to complete must look easy rather than hard [29]. Therefore, the crown was placed inside the plexiglass tube on top of a cylinder (see S2 Fig), which elevated the crown, so that it looked like it could be reached even by 4-year-olds.

## Procedure

The procedure of Study 2 was nearly identical to the help context of Study 1. However, instead of carrying the crown to the study table themselves, on the second test trial, children walked beside E1 as she carried the crown to the study table. This ensured that children's posture on

the second test trial was not altered because they were carrying an object (the crown). Additional coding confirmed that the duration of children's attempt to help did not clearly differ across conditions or age groups in Study 2 (S9 Appendix).

## Coding and analysis

**Discrete emotion coding.** We conducted an additional emotion coding with the aim of identifying the discrete emotions that were elicited by children's failed attempt to help and correlating children's emotion ratings with their change in their body posture. Our aim with this emotion coding was to assess whether children predominantly expressed a negative self-conscious emotion after failing to help. Two coders, blind to the study hypotheses, independently rated the presence of four emotions, namely, shame, anger, sadness, and happiness on a scale from 1 to 5 with 1 = "does not show this emotion at all", 3 = "shows this emotion a little bit" and 5 = "shows this emotion very much" on the first test trial [see 24 for a similar coding procedure]. Ratings were conducted based on video stills (without audio) of the *Kinect* recordings (see Fig 1D). Coders were instructed to watch all the available video frames of the first test trial and provide an overall emotion rating for the entire recording. These video stills were the same ones that were used to extract children's body posture data. Coders were provided a feature-based description of each emotion, and were instructed, in addition, to rely on their naïve intuition regarding what these emotions look like. This coding represents an improvement over the emotion coding of Study 1, because coders rated children's emotional response based on a set of objectifiable features. More specifically, we provided coders established cues to identify shame [an averted or downturned gaze, a lowered upper body posture, frowning, and a similarity to embarrassment; 29, 30, 56]. Coders were also provided with established cues to identify the three other emotions (see Table A in S9 Appendix). Inter-rater agreement was as follows: shame, $ICC = .73$ ($r = .59$), sadness, $ICC = .79$ ($r = .67$), anger, $ICC = .43$ ($r = .34$), and happiness, $ICC = .88$ ($r = .78$). We did not include guilt in our coding scheme, because guilt has been argued to have no clear nonverbal emotion display [3, 9].

**Language coding.** In Study 2, we also coded children's language to explore whether children showed evidence of explicit self-evaluation in response to failing to help [28–30, see S9 Appendix].

**Body posture analysis.** For the body posture analyses, we employed the same modeling approach as in Study 1. The primary analysis focused on the first test trial, and the models included the same random effects as in Study 1. In addition, like in Study 1, we ran omnibus models that encompassed data from both the first and second test trial.

Moreover, as preregistered, we ran an analysis in which we included age as a continuous predictor. In an exploratory analysis, we also examined the results including age as a categorical variable (4 or 5 years old), which is comparable to the analysis of developmental trends in previous studies of children's self-conscious emotions [21, 30].

## Results

### Body posture

**Change in chest expansion.** *First trial analysis*. On the first test trial, 5-year-olds' chest expansion (in cm) was below its baseline level in both the observed, 95% CI [-0.44, -2.71], and in the unobserved conditions, 95% CI [-0.05, -2.40]. Four-year-olds' chest expansion, however, did not reliably differ from baseline in either the observed, 95% CI [1.45, -0.75], or the unobserved condition, 95% CI [1.45, -1.82]. The model predicting the change in children's chest expansion on the first test trial revealed no two-way interaction of observation and age, $\chi^2(1) = 2.11$, $p = .15$, nor a three-way interaction of observation, age, and distance, $\chi^2(1) = 0.2$ $p = .66$.

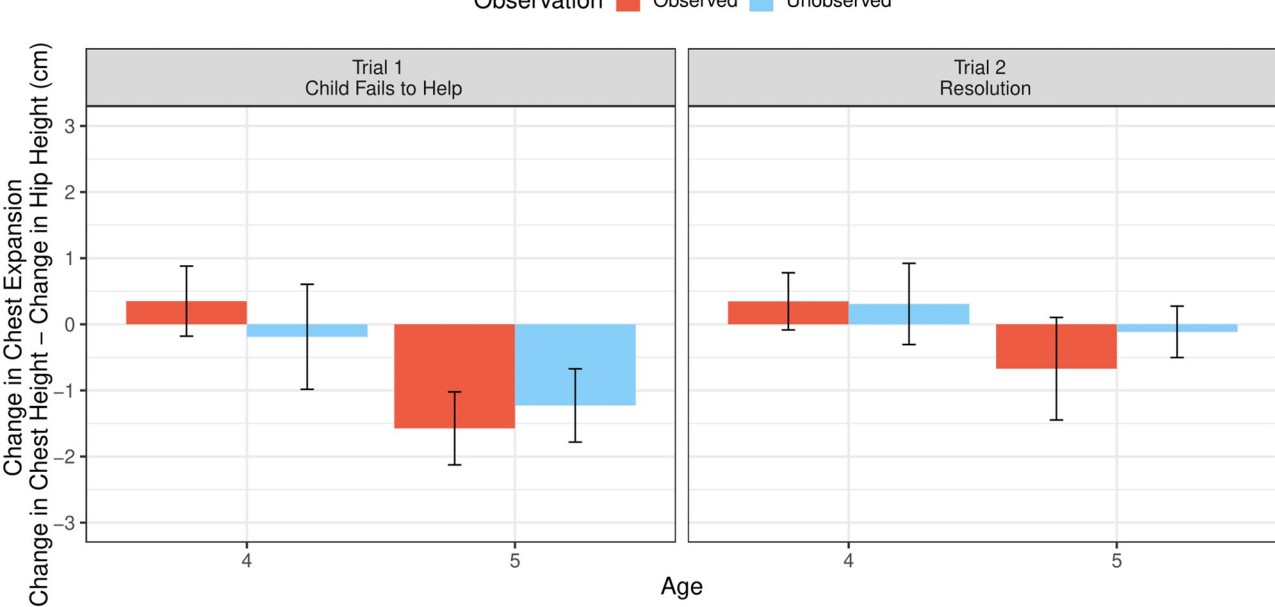

**Fig 5. Children's baseline-corrected upper body posture on the first and second test trial of Study 2.** Bars represent means and error bars represent ± 1 standard error.

There were also no overall effects of observation, $\chi^2(1) = 0$, $p = .98$, or gender, $\chi^2(1) = 0.01$, $p = .93$, on the change in children's chest expansion. The continuous age variable did not predict children's change in chest expansion on the first test trial, $\chi^2(1) = 0.81$, $p = .37$. However, in an exploratory analysis, which included age as a factor (splitting children into groups of 4- or 5-year-olds), children's change in chest expansion on the first trial varied as a function of age, $\chi^2(1) = 5.61$, $p = .02$ (see Fig 5). Five-year-olds ($M = -1.44$ cm, $SD = 2.68$) showed a greater reduction in chest expansion (in cm) after failing to help than did 4-year-olds ($M = 0.06$, $SD = 3.4$).

*Omnibus analysis (first and second trial)*. In the omnibus model, there was an influence of trial, $\chi^2(1) = 4.23$, $p = .04$ (see Fig 5). Like in Study 1, children's chest expansion (in cm) showed a greater reduction on the first test trial, immediately after children's failed attempt to help ($M = -0.67$, $SD = 3.15$), compared to the second test trial, during the resolution of the situation ($M = -0.06$, $SD = 2.75$). The continuous age variable did not predict children's change in chest expansion in the omnibus model, $\chi^2(1) = 1.12$, $p = .29$. In the exploratory analysis, which again included age as categorical variable, age was significant, $\chi^2(1) = 4.59$, $p = .03$. Averaging across both test trials, 5-year-olds ($M = -1.01$, $SD = 2.88$) showed a more reduced chest expansion (in cm) than 4-year-olds ($M = 0.18$, $SD = 2.97$).

### Discrete emotion coding

An omnibus Friedman test indicated that the type of emotion predicted children's average emotion rating on the first test trial, $\chi^2(2) = 62.7$, $p < .001$. Follow-up focused comparisons using Wilcoxon signed-rank tests with a Bonferroni-corrected alpha-level ($\alpha = 0.05/6 = .0083$) were conducted. These analyses showed that children's emotional response was rated as more shame-like than like sadness, $T = 639$, $p < .001$, 95% CI [-0.75, -0.25], $d = 0.39$; happiness, $T = 475$, $p < .001$, [-1.5, -1], $d = 1.19$; and anger, $T = 104.5$, $p < .001$, [-1.5, -1], $d = 1.46$. Similarly, children's emotional expression on the first test trial was rated as more like sadness than

**Table 1. Descriptive statistics, the results of pairwise comparisons, and results of spearman's rank-order correlations with children's change in chest expansion for the discrete emotion coding of Study 2.**

| Emotion | M (SD) | rho | p |
|---|---|---|---|
| Shame | 2.19 (0.87)$_{abd}$ | -.15 | .14 |
| Sadness | 1.88 (0.68)$_{ab}$ | -.09 | .39 |
| Anger | 1.20 (0.41)$_{a}$ | -.25 | .07 |
| Happiness | 1.28 (0.66)$_{a}$ | .25 | .01 |

Coders only rated the presence of each emotion on the first test trial (after the child fails to help). Means with different subscripts differ from each other at p < .001. Ratings were conducted on a scale from 1 = "does not show this emotion at all", 3 = "shows this emotion a little bit" to 5 = "shows this emotion a lot". Note that the ratings for shame, anger, and sadness were conducted such that higher ratings indicate more negative emotions, while for happiness higher ratings indicate a more positive emotion.

happiness, $T = 2410.5$, $p < .001$, [-1, -0.75], $d = 0.9$; and anger, $T = 323$, $p < .001$, [-1, -0.75], $d = 1.21$. There was no clear difference between the ratings of happiness and anger, $T = 499.5$, $p = .84$, [-0.75, 0.50], $d = -0.14$ (see Table 1).

Children's rating of happiness, moreover, was correlated with their change in chest expansion, spearman's $rho = .25$, $p = .01$, on the first trial. None of the other emotion ratings for negative emotions (shame, sadness, or anger) showed a similarly clear association with the change in children's chest expansion (see Table 1).

## Discussion of Study 2

The main finding of Study 2 is that children, like in Study 1, showed a reduction in upper body posture after they were unable to help. Children's upper body posture was lower immediately after they could not help—on the first test trial—than during the resolution of the situation—on the second test trial. Moreover, 5-year-olds' upper body posture was below baseline on the first test trial, thus replicating two findings of Study 1. In addition, like in Study 1, observation did not affect 4- to 5-year-old children's reduction in upper body posture. An exploratory analysis, in which children were split into two age groups (4- or 5-year-olds), however, did show an influence of age, such that 5-year-olds' upper body posture was more reduced than 4-year-olds'. We did not find the same effect of gender on children's change in upper body posture in Study 2 that we found in Study 1. It does seem that this effect in Study 1 may have been spurious, because it was not corroborated by the emotion valence coding of Study 1.

In addition, the discrete emotion coding of Study 2 revealed that children expressed more shame than sadness or anger after failing to help. Thus, the emotion coding suggests that the expression of self-conscious emotions was predominant after 4- to 5-year-old children failed to help.

Additional correlational analyses indicated that children who expressed more happiness showed a greater increase in upper body posture on the first test trial. The correlational analyses with ratings of shame, sadness and anger did not reach significance. Therefore, we conclude that changes in upper body posture are indicative of the valence of children's emotional response [see also 17, 45, 46] while children in the current study showed additional features of emotion that allowed raters to identify children's emotion as similar to shame (and to a lesser extent similar to sadness). It is important to note that while happiness was the only emotion that showed a clear association with the change in children's upper body posture in the current study, this does not mean that changes in upper body posture can only be reflective of happiness. We think a

larger study exploring children's emotional responses across a more diverse set of tasks is needed to clarify which kinds of emotions can be reflected through changes in upper body posture.

## General discussion

The current studies represent the first investigation of children's emotional response to failing to help others using a method that automatically and objectively record changes in children's body posture. Our studies show that young children's emotional response is similarly negative when they fail to help or fail to achieve their own goal in both an observed and unobserved setting. Specifically, in both studies, children expressed a greater reduction in upper body posture after they failed to help (Trial 1) than during the resolution of the situation moments later (Trial 2). This result was corroborated by the emotion valence coding of Study 1. While observation or goal context did not influence this emotional response, we did find evidence in Study 2 that 5-year-olds expressed a greater reduction in upper body posture after failing to help than 4-year-olds. Moreover, in Study 2, children expressed a predominantly shame-like negative emotion after failing to help, suggesting that self-evaluative processes were involved in children's emotional response.

### The influence of observation

Children expressed similarly negative emotions regardless of whether they were observed or unobserved during a failure to help, suggesting that the presence of an audience is not required for young children to express a negative self-conscious emotion. It is worth noting that children were made aware of the observer's presence twice during the studies and were told that the observer would watch them today, which is comparable to previous studies of the influence of observation on children's prosocial behavior [40, 41]. Our findings thus raise questions about the role of others' evaluation or judgment of oneself in young children's expression of self-conscious emotions. Some scholars have argued that young children's expression of shame following achievement-related failures is the result of observing adults knowing (or having the impression) that children have performed poorly until children are school-aged [28, 38]. In observed settings, children's expression of self-conscious emotions could function to appease onlookers who are aware of children's poor behavior or who know that they fell short of a social standard [3, 35, 36, 47]. Other scholars have maintained that self-conscious emotions are the result of internal self-evaluations that emerge in toddlerhood and early childhood [30, 39]. Our findings align with the latter conception of self-conscious emotions, by suggesting that children may have engaged in negative *self*-evaluations after failing to help and after failing to complete their own goal, rather than focusing the observers' judgements of themselves. Thus others' evaluations may feature in the expression of negative self-conscious emotions, but rather because norms (as expressed in others' evaluation) regarding what one ought to do become internalized, and children begin to evaluate themselves and their behavior from the perspective their social group [25].

We note that children were not explicitly asked whether they evaluated themselves negatively after failing to help. Therefore, it is possible that children expressed a negative emotion because they feared punishment through the experimenter (E1) once she returned. This interpretation of children's emotional response appears unlikely, however. Children were led to believe that E1 would not return, and that their failure to help would therefore go unnoticed by the helpee.

### Age-related developments

Failing to help others lowered 5-year-olds' upper body posture more than 4-year-olds'. It must be pointed out that this finding was not predicted, and only emerged in exploratory analyses.

There nevertheless may be differences in the degree to which 4- and 5-year-olds felt that helping was required of them, which influenced children's emotional response. A developmental shift in children's normative expectations of helping aligns with previous work showing that 5-to 6-year-olds, but not 3- to 4-year-olds, protest third parties' failures to be charitable [31]. In line with this investigation, older children in the current studies might have felt more like they ought to have helped. Relatedly, Vaish, Carpenter and Tomasello [22] documented a developmental increase in children's distinctly guilt-like responses between 2 and 3 years of age. A similar age-related increase in the expression of shame has been found in young children's emotional responses to achievement-related failures [26, 27]. Furthermore, one comprehensive analysis of children's emotional expression in response to a mishap found that bodily expressions of tension increased between 2 to 4 years of age [21]. Therefore, the finding that 5-year-olds show a greater decrease in upper body posture after failing to help than 4-year-olds, in part, aligns with a body of work suggesting that children, with age, respond with more negative emotions to their own transgressions or failures to meet social standards. However, at present, there is a lack of studies on the development of children's negative self-conscious emotions across a wide age range in early childhood. Therefore, more research is needed to explore the developmental roots of children's negative self-conscious emotions after failing to help and in response to other mishaps, transgressions, and violations of (pro)social norms.

Another, more methodological, question raised by the present study is why our analysis of children's change in upper body posture showed an effect of age when age was coded as a categorical variable, but not when age was coded as a continuous predictor. Indeed, in our preregistration we decided to include age as a continuous predictor in our models because we expected increased power to detect effects of interest when using age as a covariate. We could only speculate as to why this pattern of results emerged. We think the more important conclusion is that no effects of age were predicted; therefore, in our view, the finding that age appears to influence children's emotional response requires further replication in a study that is adequately powered to detect such effects.

## Children's emotional responses to failures to help and failures to complete their own goals

Children expressed similar emotions in both the own-goal and help contexts. Thus, we cannot conclude that children's emotional response is specific to being unable to be prosocial. This finding raises the question to what extent a concern for others' needs is the driving motive underlying young children's helping [e.g., 57, 58]. The most other-oriented response in the current studies would have been one in which children express a more negative emotion when they fail to help compared to when they fail to achieve their own goals. In our study, children may have treated the own-goal and the help goal contexts similarly, because they both involved a standard that children attempted to achieve (helping or completing ones' own tower). As pointed out by Eisenberg, VanSchyndel, and Spinrad [59], although these are partially self-oriented, achievement-related motives (e.g., feelings of competence and self-esteem or the adherence to internalized moral norms) nevertheless may motivate prosocial behavior in young children. Thus, our findings suggest that achievement-related motives, similar to the motives that children have when they complete their own goals, are part of children's motivation to help others.

It is also possible that children adopted the goal of completing the tower as their own in both the help and the own-goal contexts, and that children's emotional response therefore was always the result of not completing a goal that was established by an experimenter (see 58 for a discussion of goal-alignment models of early helping behavior). We cannot rule out this

possibility. Indeed, this possibility is consistent with children possessing achievement-related motivations to complete the action in both the own-goal and the help context. Yet, while achievement-related concerns may be the central motive underlying the fulfillment of children's own goals, there may, in some contexts, be additional motivations underlying children's helping [e.g., a concern for others' needs; 57, 58].

Another possibility is that additional training and instructions regarding the different outcomes would have further facilitated children's understanding of the own-goal and help contexts. Similarly, additional comprehension checks could have been included to assess whether children fully grasped that their actions would fulfil their own goal or the goal of the experimenter. Such improvements notwithstanding, we do think that children noticed the difference between the two contexts for the following reason. In the help context, the experimenter referred to her own need for the tower to be completed several times, while there was no mention of the experimenter's needs in the own-goal context (see S3 and S7 Appendices). Nevertheless, it remains an important question for future investigations to assess to what extent children understood that their helping actions fulfill others' needs as compared to fulfilling a goal that children have themselves [see also 57–59 for a more detailed discussion of multiple motivations that may underlie early helping behavior, some of which do not require an understanding that another's needs would be fulfilled].

In future studies, it will be worthwhile to investigate whether and when children express negative emotions after being unable to behave prosocially in a costlier context, that is when children's own goal and the goal of helping others are not aligned. Such an investigation could reveal when children value being prosocial (e.g., by sharing resources with others) more than achieving their own goals [e.g., by receiving a reward for themselves; see for instance the costly helping context of 60, 61]. In these contexts, the fulfillment of children's own material desires would contrast more clearly with the fulfillment someone else's material needs than in the current study using a rather low-cost helping task.

## Limitations and future directions

Our studies show that children expressed a negative emotional response that is predominantly like shame (i.e., a self-conscious emotion) in response to failing to help. Importantly, while this effect was small to moderate, children in Study 2 also showed more shame than sadness according to the discrete emotion coding we conducted. The coding scheme provided to raters used objectifiable features of each emotion, which were themselves based on prior work (see Table A in S9 Appendix). It remains an open question, nevertheless, whether our body posture measures would be able to differentiate more shame-like emotions as compared to children's emotional expression of sadness, anger, or frustration. Future work could, for instance, vary if children merely witness a failed attempt to help or themselves fail to help someone and measure children's body posture thereafter. While the former context should cause sadness or sympathy with the person whose goal remained unfulfilled [43, 44, 57], the latter context should cause shame- or guilt-like emotion, because children are directly responsible for someone else's negative outcome [see 22 for a similar study]. Perceived causal responsibility for harm to others has been argued to be a critical antecedent to the elicitation of shame or guilt [22, 62]. Measuring children's body posture in such a setting could reveal if sadness or sympathy versus self-conscious emotions elicit different bodily emotional expressions.

Additionally, since our aim was to investigate if and when children expressed a self-conscious emotion through their nonverbal emotion display, we did not assess whether young children express additional features of self-conscious emotions after failing to help, such as reparative prosocial behavior towards the person who was not helped [19, 20, 22]. Some

scholars have argued that children's reparative prosocial behaviors (including apologies) are indicative of guilt, while young children express shame predominantly through nonverbal behaviors, such as gaze aversions, a tense, hunched or lowered posture and an avoidance of the person harmed [19, 20, 23, 29]. Adding a direct assessment of children's change in body posture to the analysis of other shame- and guilt-relevant behaviors could shed new light on whether these emotional responses are indeed distinct in early childhood.

## Conclusion

Young children expressed negative emotions (as measured via observational coding and through a reduction in their upper body posture) after failing to help others and after failing to complete their own goal both when they were observed and when they are unobserved. An exploratory analysis showed that 5-year-olds expressed a greater reduction in upper body posture than 4-year-olds. We conclude that young children's emotional expression indicates that extent to which children feel that helping others is personally binding. Thereby our findings suggest that preschool children develop a concern with helping others that is, in part, motivated by (the anticipation of) negative emotions, like shame or guilt, after failing to help.

## Supporting information

**S1 Fig. An overview of the study set-up from a bird's-eye perspective.**
(DOCX)

**S2 Fig. An illustration of the materials of Studies 1 and 2.**
(DOCX)

**S1 Appendix. Recording and pre-processing of body posture data (Studies 1 and 2).**
(DOCX)

**S2 Appendix. Additional details regarding the participants of Study 1.**
(DOCX)

**S3 Appendix. Detailed procedure of Study 1.**
(DOCX)

**S4 Appendix. Additional body posture results of Study 1.**
(DOCX)

**S5 Appendix. Additional coding for Study 1.**
(DOCX)

**S6 Appendix. Additional details regarding the participants of Study 2.**
(DOCX)

**S7 Appendix. Additional details regarding the materials and procedure of Study 2.**
(DOCX)

**S8 Appendix. Additional body posture results for Study 2.**
(DOCX)

**S9 Appendix. Additional coding for Study 2.**
(DOCX)

## Acknowledgments

We thank all of the research assistants who helped with recruiting, data collection and coding. In particular, we thank Katja Kirsche, Sina Gibhardt, Özkan Cavus, Gökhan Sürer, Johanna Seumel, Alexander Porto, Anneke Elsner, Jasmin Biber, Souromi Bhouwik, and Rabia Hassan for their help with data collection. We in addition thank Jasmin Biber, Alina Kuzima, Leonore Blume, Anna-Lina Rauschenbach, Selma Kahlhorn, and Christina Kellermann for their help with coding. Special thanks to Elmar Tarajan and Anja Neumann for writing the scripts to run the Kinect and to Kim-Laura Speck for help with the Matlab code.

## Author Contributions

**Conceptualization:** Stella C. Gerdemann, Jenny Tippmann, Bianca Dietrich, Jan M. Engelmann, Robert Hepach.

**Data curation:** Stella C. Gerdemann.

**Formal analysis:** Stella C. Gerdemann.

**Funding acquisition:** Robert Hepach.

**Investigation:** Stella C. Gerdemann, Jenny Tippmann, Bianca Dietrich.

**Methodology:** Stella C. Gerdemann.

**Project administration:** Stella C. Gerdemann, Jenny Tippmann.

**Resources:** Robert Hepach.

**Supervision:** Jan M. Engelmann, Robert Hepach.

**Visualization:** Stella C. Gerdemann.

**Writing – original draft:** Stella C. Gerdemann.

**Writing – review & editing:** Jenny Tippmann, Bianca Dietrich, Jan M. Engelmann, Robert Hepach.

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
