## [Decision Letter · Decision Letter 0]

25 Nov 2021

PONE-D-21-34250Young children show negative emotions after failing to help othersPLOS ONE

Dear Dr. Gerdemann,

Thank you for submitting your manuscript to PLOS ONE. Two reviewers have carefully read this paper, as have I. As you'll see, both reviewers expressed considerable enthusiasm about this paper. I share their enthusiasm, and agree that this paper is innovative, exciting, and technically sound. The reviewers have only minor comments that need addressing, and therefore I am inviting you to submit a revised version of the manuscript that addresses these points.

We look forward to receiving your revised manuscript.

Kind regards,

Lucas Payne Butler

Academic Editor

PLOS ONE

Journal Requirements:

(Research for this project was partly funded by a ‘Flexible Fund’ grant awarded to R.H.)

(Research for this project was partly funded by a ‘Flexible Fund’ grant awarded to R.H.)

(Research for this project was partly funded by a ‘Flexible Fund’ grant awarded to R.H.)

6. We note that Figure 1 includes an image of a participant in the study. 

Reviewers' comments:

Reviewer's Responses to Questions

**Comments to the Author**

1. Is the manuscript technically sound, and do the data support the conclusions?

Reviewer #1: Yes

Reviewer #2: Yes

2. Has the statistical analysis been performed appropriately and rigorously? 

Reviewer #1: Yes

Reviewer #2: Yes

3. Have the authors made all data underlying the findings in their manuscript fully available?

Reviewer #1: Yes

Reviewer #2: Yes

4. Is the manuscript presented in an intelligible fashion and written in standard English?

Reviewer #1: Yes

Reviewer #2: Yes

5. Review Comments to the Author

Reviewer #1: The current study found that 5-year-olds in Germany show negative emotional responses after failing to help another person. Whereas, 4-year-olds were less likely to show negative emotional responses. Importantly, children’s negative emotional responses did not differ significantly by whether they are being observed or whether the completion of the task was children’s own goal or another person’s goal. Negative emotional responses were measured by a reduce in children’s chest expansion, which is a creative and novel contribution to the existing literature. Further, the results were strengthened by converging evidence from additional emotion coding.

This paper is well-written with clear theoretical motivations. Also, I would like to applaud the authors for their investment of time and effort to the novel paradigm, i.e., body posture recording. I only have minor suggestions/questions for clarifications.

1. In General Discussion, the authors discussed the possibility that children did not distinguish E1’s goal and their own goal (line 851 on p.34). I agree with this possibility as the distinction may not be clear to children. One way to address this in the future research could be to introduce a reward in a task. For example, a child participant receives stickers when they complete a task (child’s own goal condition) or helpee receives stickers when children help the person complete the task (helpee’s goal condition). This way, it could be clearer for children how their helping benefits themselves vs. other. I do not request an additional experiment for this paper, but it may be worth noting this as a future direction in General Discussion.

2. In some places, the authors’ names were omitted. Instead, it read e.g., “[27] found that preschool children..” or “In addition, [37] reported the results..” on page 7. I wasn’t sure if this is due to the journal’s policy or the authors’ mistake in editing. But I just wanted to bring this into attention in case this is a mistake, not a part of the journal’s guidelines.

3. It was confusing that age as a factor (4yos vs. 5yos), but not age as a continuous variable, predicted changes in chest expansion (line 696 on p. 27). From my understanding, age in months is more fine-grained (i.e., high resolution), allowing us to detect an age effect (if any). How can readers reconcile seemingly inconsistent results from the two analyses (one with no age effects and the other with an age effect)? That is, what did the inconsistency occur?

Reviewer #2: Before starting my review, I would like to note that I had previously reviewed an earlier version this article for another journal. In the previous version of this manuscript, my main recommendation was adding a final concluding paragraph to tie everything together, and this change has been made in the current version of the manuscript. Therefore, my overall enthusiasm for the paper continues to be very high.

Summary: The present paper finds that young children upon not being able to help others show a lower upper body posture. Furthermore, 5-year-olds showed even lower upper body posture than 4-year-olds after failing to help others. Importantly, changes in children’s body posture do not seem to be affected by whether children were observed or not during the session. These findings are interpreted in the paper as children experiencing negative emotions upon not being able to be prosocial.

I would like to highlight that this is an important and novel paper that will greatly benefit emotion and prosociality researchers from both developmental and social psychology backgrounds. The paper uses novel methods, namely body posture analyses to investigate whether children show self-conscious emotions (physically) after failing to help others. Despite the relative novelty of the methods, all studies have been preregistered and the paper is very forthcoming with deviations from the planned analyses. Based on the theoretical and practical reasons provided in the manuscript, these deviations and exploratory analyses are very reasonable and supported by the literature.

6. PLOS authors have the option to publish the peer review history of their article (what does this mean?). If published, this will include your full peer review and any attached files.

Reviewer #1: No

Reviewer #2: No

---

## [Author Response · Author response to Decision Letter 0]

22 Dec 2021

Revision of our manuscript entitled “Young children show negative emotions after failing to help others” (PONE-D-21-34250)

Dear Dr. Butler, 

You wrote: 

 “Thank you for submitting your manuscript to PLOS ONE. Two reviewers have carefully read this paper, as have I. As you'll see, both reviewers expressed considerable enthusiasm about this paper. I share their enthusiasm, and agree that this paper is innovative, exciting, and technically sound. The reviewers have only minor comments that need addressing, and therefore I am inviting you to submit a revised version of the manuscript that addresses these points.”

Response: 

Thank you for the positive assessment of our work! Below we provide our revisions in response to the points raised by the reviewers in a point-by-point manner. 

Reviewer #1: 

(1)

“The current study found that 5-year-olds in Germany show negative emotional responses after failing to help another person. Whereas, 4-year-olds were less likely to show negative emotional responses. Importantly, children’s negative emotional responses did not differ significantly by whether they are being observed or whether the completion of the task was children’s own goal or another person’s goal. Negative emotional responses were measured by a reduce in children’s chest expansion, which is a creative and novel contribution to the existing literature. Further, the results were strengthened by converging evidence from additional emotion coding.

This paper is well-written with clear theoretical motivations. Also, I would like to applaud the authors for their investment of time and effort to the novel paradigm, i.e., body posture recording. I only have minor suggestions/questions for clarifications.”

Response: Thank you for the positive assessment of our work. We have enclosed our responses to the points raised below. 

(2)

“In General Discussion, the authors discussed the possibility that children did not distinguish E1’s goal and their own goal (line 851 on p.34). I agree with this possibility as the distinction may not be clear to children. One way to address this in the future research could be to introduce a reward in a task. For example, a child participant receives stickers when they complete a task (child’s own goal condition) or helpee receives stickers when children help the person complete the task (helpee’s goal condition). This way, it could be clearer for children how their helping benefits themselves vs. other. I do not request an additional experiment for this paper, but it may be worth noting this as a future direction in General Discussion.”

Response: This is an interesting point and we have added several clarifications to the General Discussion in response. 

First, we clarify that the notion that children may have adopted the goal of the experimenter as their own in the help context aligns with previous goal-alignment models of early helping behavior. We also point out that this does not contradict our main interpretation of the results across the two goal contexts, namely that achievement-related motives, in part, can explain children’s motivation to help others (p. 35, l. 874-882): 

It is also possible that children adopted the goal of completing the tower as their own in both the help and the own-goal contexts, and that children’s emotional response therefore was always the result of not completing a goal that was established by an experimenter (see 58 for a discussion of goal-alignment models of early helping behavior). We cannot rule out this possibility. Indeed, this possibility is consistent with children possessing achievement-related motivations to complete the action in both the own-goal and the help context. Yet, while achievement-related concerns may be the central motive underlying the fulfillment of children’s own goals, there may, in some contexts, be additional motivations underlying children’s helping (e.g., a concern for others’ needs; 57,58).

In addition, we have added a paragraph discussing whether children may not have understood the distinction between the help and the own-goal context and illustrate ways in which future work could assess whether children grasped this distinction (p. 35, l. 883 – p. 36, l. 895):

Another possibility is that additional training and instructions regarding the different outcomes would have further facilitated children’s understanding of the own-goal and help contexts. Similarly, additional comprehension checks could have been included to assess whether children fully grasped that their actions would fulfil their own goal or the goal of the experimenter. Such improvements notwithstanding, we do think that children noticed the difference between the two contexts for the following reason. In the help context, the experimenter referred to her own need for the tower to be completed several times, while there was no mention of the experimenter’s needs in the own-goal context (see S3 and S7 Appendix). Nevertheless, it remains an important question for future investigations to assess to what extent children understood that their helping actions fulfill others’ needs as compared to fulfilling a goal that children have themselves (see also 57–59 for a more detailed discussion of multiple motivations that may underlie early helping behavior, some of which do not require an understanding that another's needs would be fulfilled).

Finally, we agree that measuring children’s emotional response in a context in which there is a more clearly defined difference in the outcome depending on whether the child’s or someone else’s goal is fulfilled would be very interesting. As the reviewer mentions, such a context could be established if either the child or someone else receives a reward depending on whose goal is fulfilled (similar to a costly helping context). We now describe a paradigm like the one mentioned by the reviewer in our General Discussion (p. l. 36, l. 896-903): 

In future studies, it will be worthwhile to investigate whether and when children express negative emotions after being unable to behave prosocially in a costlier context, that is when children’s own goal and the goal of helping others are not aligned. Such an investigation could reveal when children value being prosocial (e.g., by sharing resources with others) more than achieving their own goals (e.g., by receiving a reward for themselves; see for instance the costly helping context of 60,61). In these contexts, the fulfillment of children’s own material desires would contrast more clearly with the fulfillment someone else’s material needs than in the current study using a rather low-cost helping task. 

(3)

In some places, the authors’ names were omitted. Instead, it read e.g., “[27] found that preschool children..” or “In addition, [37] reported the results..” on page 7. I wasn’t sure if this is due to the journal’s policy or the authors’ mistake in editing. But I just wanted to bring this into attention in case this is a mistake, not a part of the journal’s guidelines.

Response: Thank you for pointing out that some of the citations did not adhere to the journal’s reference style guidelines. We have edited the in-text references throughout our manuscript. 

For instance, the above passages now read as follows (p. 7, l. 161-163 and p. 7, l. 173-175): 

In one study, Holodynski found that preschool children between 3 to 6 years of age are more likely to show shame (according to the author’s definition) when they fail a task while they are observed than while they are alone (28).

In addition, Harter reported the results of interview-studies showing that children at age 5 to 6 only mentioned whether a parent was ashamed of them in response to hypothetical situations in which they failed a task (during an athletic competition; 38).

(4)

It was confusing that age as a factor (4yos vs. 5yos), but not age as a continuous variable, predicted changes in chest expansion (line 696 on p. 27). From my understanding, age in months is more fine-grained (i.e., high resolution), allowing us to detect an age effect (if any). How can readers reconcile seemingly inconsistent results from the two analyses (one with no age effects and the other with an age effect)? That is, what did the inconsistency occur?

Response: We understand that the results are not entirely consistent across different methods of encoding age. At present we could only speculate as to why this pattern of results emerged. We would point out that either way any effect of age was not predicted; therefore, the exploratory analysis using age as a categorical variable is primarily meant to inform future studies. We now point this out in the General Discussion section (p. 34, l. 849-857): 

Another, more methodological, question raised by the present study is why our analysis of children’s change in upper body posture showed an effect of age when age was coded as a categorical variable, but not when age was coded as a continuous predictor. Indeed, in our preregistration we decided to include age as a continuous predictor in our models because we expected increased power to detect effects of interest when using age as a covariate. We could only speculate as to why this pattern of results emerged. We think the more important conclusion is that no effects of age were predicted; therefore, in our view, the finding that age appears to influence children’s emotional response requires further replication in a study that is adequately powered to detect such effects.

Reviewer #2: 

(1)

Before starting my review, I would like to note that I had previously reviewed an earlier version this article for another journal. In the previous version of this manuscript, my main recommendation was adding a final concluding paragraph to tie everything together, and this change has been made in the current version of the manuscript. Therefore, my overall enthusiasm for the paper continues to be very high.

Summary: The present paper finds that young children upon not being able to help others show a lower upper body posture. Furthermore, 5-year-olds showed even lower upper body posture than 4-year-olds after failing to help others. Importantly, changes in children’s body posture do not seem to be affected by whether children were observed or not during the session. These findings are interpreted in the paper as children experiencing negative emotions upon not being able to be prosocial.

I would like to highlight that this is an important and novel paper that will greatly benefit emotion and prosociality researchers from both developmental and social psychology backgrounds. The paper uses novel methods, namely body posture analyses to investigate whether children show self-conscious emotions (physically) after failing to help others. Despite the relative novelty of the methods, all studies have been preregistered and the paper is very forthcoming with deviations from the planned analyses. Based on the theoretical and practical reasons provided in the manuscript, these deviations and exploratory analyses are very reasonable and supported by the literature.

Response: Thank you for the positive assessment of our work! Following the reviewer’s suggestion in a review of this paper for another journal, we added a concluding paragraph to the General Discussion (p. 37, l. 933-p. 38, l. 941):

Young children expressed negative emotions (as measured via observational coding and through a reduction in their upper body posture) after failing to help others and after failing to complete their own goal both when they were observed and when they are unobserved. An exploratory analysis showed that 5-year-olds expressed a greater reduction in upper body posture than 4-year-olds. We conclude that young children’s emotional expression indicates that extent to which children feel that helping others is personally binding. Thereby our findings suggest that preschool children develop a concern with helping others that is, in part, motivated by (the anticipation of) negative emotions, like shame or guilt, after failing to help.

---

## [Decision Letter · Decision Letter 1]

23 Mar 2022

Young children show negative emotions after failing to help others

PONE-D-21-34250R1

Dear Dr. Gerdemann,

We’re pleased to inform you that your manuscript has been judged scientifically suitable for publication and will be formally accepted for publication once it meets all outstanding technical requirements.

Kind regards,

Alexandra Paxton

Academic Editor

PLOS ONE

Additional Editor Comments (optional):

Reviewers' comments:

Reviewer's Responses to Questions

**Comments to the Author**

1. If the authors have adequately addressed your comments raised in a previous round of review and you feel that this manuscript is now acceptable for publication, you may indicate that here to bypass the “Comments to the Author” section, enter your conflict of interest statement in the “Confidential to Editor” section, and submit your "Accept" recommendation.

Reviewer #1: All comments have been addressed

Reviewer #2: All comments have been addressed

2. Is the manuscript technically sound, and do the data support the conclusions?

Reviewer #1: Yes

Reviewer #2: Yes

3. Has the statistical analysis been performed appropriately and rigorously? 

Reviewer #1: Yes

Reviewer #2: Yes

4. Have the authors made all data underlying the findings in their manuscript fully available?

Reviewer #1: Yes

Reviewer #2: Yes

5. Is the manuscript presented in an intelligible fashion and written in standard English?

Reviewer #1: Yes

Reviewer #2: Yes

6. Review Comments to the Author

Reviewer #1: The authors have skillfully addressed my comments and questions about their manuscript. I believe it will make a meaningful contribution to the literature.

Reviewer #2: The authors were very responsive to all of my comments. The current manuscript is much improved and I have no further suggestions.

7. PLOS authors have the option to publish the peer review history of their article (what does this mean?). If published, this will include your full peer review and any attached files.

Reviewer #1: No

Reviewer #2: No

---

## [Editor Report · Acceptance letter]

29 Mar 2022

PONE-D-21-34250R1 

Young children show negative emotions after failing to help others 

Dear Dr. Gerdemann:

I'm pleased to inform you that your manuscript has been deemed suitable for publication in PLOS ONE. Congratulations! Your manuscript is now with our production department. 

Kind regards, 

on behalf of

Dr. Alexandra Paxton 

Academic Editor

PLOS ONE